# Potential Drug–Nutrient Interactions of 45 Vitamins, Minerals, Trace Elements, and Associated Dietary Compounds with Acetylsalicylic Acid and Warfarin—A Review of the Literature

**DOI:** 10.3390/nu16070950

**Published:** 2024-03-26

**Authors:** David Renaud, Alexander Höller, Miriam Michel

**Affiliations:** 1DIU MAPS, Fundamental and Biomedical Sciences, Paris-Cité University, 75006 Paris, France; 2DIU MAPS, Health Sciences Faculty, Universidad Europea Miguel de Cervantes, 47012 Valladolid, Spain; 3Fundacja Recover, 05-124 Skrzeszew, Poland; 4Department of Nutrition and Dietetics, University Hospital Innsbruck, 6020 Innsbruck, Austria; 5Department of Child and Adolescent Health, Division of Pediatrics III—Cardiology, Pulmonology, Allergology and Cystic Fibrosis, Medical University of Innsbruck, 6020 Innsbruck, Austria

**Keywords:** acetylsalicylic acid, warfarin, micronutrient, deficiency, potentiation, drug–nutrient interaction, triage theory

## Abstract

In cardiology, acetylsalicylic acid (ASA) and warfarin are among the most commonly used prophylactic therapies against thromboembolic events. Drug–drug interactions are generally well-known. Less known are the drug–nutrient interactions (DNIs), impeding drug absorption and altering micronutritional status. ASA and warfarin might influence the micronutritional status of patients through different mechanisms such as binding or modification of binding properties of ligands, absorption, transport, cellular use or concentration, or excretion. Our article reviews the drug–nutrient interactions that alter micronutritional status. Some of these mechanisms could be investigated with the aim to potentiate the drug effects. DNIs are seen occasionally in ASA and warfarin and could be managed through simple strategies such as risk stratification of DNIs on an individual patient basis; micronutritional status assessment as part of the medical history; extensive use of the drug–interaction probability scale to reference little-known interactions, and application of a personal, predictive, and preventive medical model using omics.

## 1. Introduction

### 1.1. Drug–Nutrient Interactions (DNIs)

Classically, pharmacological interactions (drug–drug) are considered, as well as interactions with the diet (food–drug) [1,2,3,4] or with phytotherapeutic agents (herb–drug) [5,6,7,8]. Little attention is given to DNI, which typically is 100-fold less researched than drug–drug interaction, but still clinically relevant [9,10,11,12,13]. Several classifications of DNIs are in use [12]. Medication can impact the patient’s micronutritional status—levels of essential micronutrients for proper physiological functioning—either directly or indirectly [9]. These mechanisms may be physicochemical, physiological, or pathophysiological [11,14]. There are four types of DNIs [15]: (1) bio-inactivation; (2) altered absorption type; (3) altered effect; (4) altered excretion [9,11,16,17].

Medication can directly compete with micronutrients when they use the same metabolic and transport pathways, modifying the pharmacokinetics or micronutrient status [18,19,20] (absorption, distribution, metabolism, excretion) [21]. Physiochemical interactions are mostly represented by a chelation, causing loss of a nutrient and lower drug activity. Medication can directly induce a physiological change affecting the micronutrient [10,22]. Medication can indirectly affect patient health and micronutritional status [23,24,25]. Physiological mechanisms include drug-induced changes in appetite, digestion, or renal transformation, but also drug-induced micronutritional status [23]. Pathophysiological mechanisms occur when the drug impairs nutrient uptake or usage. A typical case is when the drug alters absorption of a nutrient or when drug toxicity inhibits a metabolic process [26]. These interactions may vary in different tissues, with tissue-specific interactions [11,27]. Many DNIs are mutual—for instance, the drug affects micronutritional status, and the micronutrient affects drug metabolism, contributing to adverse drug effects (Figure 1) [28,29,30]. 

DNIs are not always negative [31,32]. As Thurnham defined, a well-balanced diet should contain ‘the adequate micronutrients to sustain body functions, but also a wide range of non-nutrients to optimize phase I (oxidation) and phase II (conjugation) metabolic processes’ [28]. These interactions can also be harvested to potentiate or reduce side effects of a medication [33]. Some drug regimens may improve micronutritional status [34,35]. DNIs can cause clinically visible (short-term) life-threatening health disorders [36] but also cause undetected long-term aging-associated health issues [37,38]. The mechanism behind age-associated chronic disease is hypothesized by the Triage theory of Bruce Ames [39]: “When micronutrient availability is limited, functions required for short-term survival and reproduction take precedence over functions whose loss can be tolerated”. This hypothesis was applied to vitamin K [40,41], with proteins required for short-term survival like blood clotting served first, and selenium, with selenoproteins [42].

### 1.2. Quality of Evidence–Clinical Speculation or Theoretical Major Burden?

A relatively small list of DNIs is documented [12]. Potential interactions have often yet to be evaluated for their incidence and clinical significance. Evaluation of drug safety is mostly about drug–drug interactions [43]. DNIs are not addressed during the drug development phase [44]. As stated in the handbook of drug–nutrient interactions [45], most data come either traced from pharmacological studies or reports from clinicians. As for pharmacological studies, DNIs are sourced from pharmacokinetic data from studies with mostly small numbers of subjects, focusing only on the clinical relevance of the interaction, with some focusing on the mechanism behind the interaction. As for clinical reports, most data come from case series and anecdotal reports. Very few studies were conducted with control groups. Some studies lack clarity on the baseline status of micronutrients, which is important for describing DNIs accurately. Many data come from animal models and cannot be extrapolated into clinical recommendations. The overall recommendation in the field of DNIs is to consider the risk of interaction on an individual patient basis. Large-scale recommendations cannot be made without larger studies. It is expected to improve over time in a similar way drug–drug interactions are now widely recognized. Historically, drug–drug interactions appeared in the early 1960s [46]. From 1970, the Swedish drug regulatory agency required pharmaceutical companies to issue annual reviews of drug interactions [46]. Knowledge on DNIs in the education of medical doctors and pharmacists is still scarce. When DNIs are included in the education of pharmacists, the recognition of their importance is growing, and students have reported encountering DNIs on a weekly basis [47].

### 1.3. Acetylsalicylic Acid (ASA)—An Overview

The antiplatelet drug ASA is the most used pharmacon worldwide [48]. The mechanism of ASA involves the inhibition of prostaglandin synthesis [49,50] through acetylation of the platelet cyclooxygenase–isozymes (COX) at functionally important amino acids (COX-1, isoform of the platelet enzyme, at serine530 [51]; COX-2, isoform at serine516) [52,53]. This prevents arachidonic acid [54] from accessing the catalytic site of the enzyme at tyrosine385 [55] and results in irreversible inhibition of cyclic prostanoids formation (platelet-dependent thromboxane A2 (TXA2), prostaglandins, and prostacyclin [53]). ASA preferably inhibits COX-1, with a 150–200 times higher affinity for COX-1 than for COX-2 [49,53]. It is debated if ASA might have antithrombotic properties [56] through modulation of clotting factors such as thrombin, fibrinogen, factor XIII, and tissue plasminogen [57]. Other mechanisms are discussed in the literature, such as changes in endothelial iron status [58]. ASA efficiency may be impacted by polymorphisms of MDRP1 and methylation status [59]. Low-dose ASA is used for both primary and secondary prevention of cardiovascular disease [60], with varying success [61]. Side effects like upper gastrointestinal injury are described in the literature [62]. 

### 1.4. Warfarin—An Overview

The anticoagulant drug warfarin is a widely used drug for prophylaxis [63]. Its effect relies on blocking the function of the vitamin K epoxide reductase complex (VKOR) in the liver, then inhibiting the K-dependent coagulation factors II, VII, IX, and X [64]. Warfarin also inhibits endogenous production of protein C and protein S [65]. The main indication of warfarin is to prevent deep-vein thrombosis, thromboembolic events, and reduce the risk of stroke in patients with atrial fibrillation, valvular disease, or an artificial heart valve [66]. The drug is a mixture of R and S-enantiomers, with the latter being the most potent. Warfarin is metabolized in the liver by the cytochrome P450 (CYP) pathway [67,68,69,70]. Among the six enzymes of the CYP family [71], four are involved in metabolizing warfarin. The S-enantiomer is mostly metabolized through the CYP2C9 enzyme to 7-hydroxywarfarin [68] and partially by CYP3A4 [69,72]. The R-enantiomer is metabolized by CYP1A2 to 6- and 8-hydroxywarfarin [68] and CYP3A4 to 10-hydroxywarfarin [68]. CYP2C8, CYP2C18, and CYP2C19 are also involved [72,73,74,75]. Most cytochrome P450 enzymes are known to be polymorphic [76], with more than 2000 variants described [77]. CYP2C9 is one of the most present CYPs in the liver, and its polymorphisms are of clinical relevance [78,79,80]. In contrast, the importance of CYP2C19 genotype variations is debated in pharmacokinetics of warfarin enantiomers [81]. VKOR and CYP polymorphisms may not be the only explanation for the dose variability of warfarin. Gut microbiota status may modulate warfarin efficiency, as it produces endogenous vitamin K [82,83]. Other genetic polymorphisms, such as CYP4F2 [84] or ABCB1 [85], are also involved. One haplotype of ABCB1 was overrepresented among patients who had a stable INR with a low dose warfarin.. Warfarin requires periodic functional monitoring, with International Normalized Ratio (INR) monitoring having become the standard [86]. INR response is known to be influenced by diet, liver function, comorbidities, drug interactions [87], as well as unpredictable patient-specific responses, partially due to polymorphisms of CYP [88,89,90,91]. 

## 2. Methods

Our literature review covers DNIs between 45 nutrients and ASA and warfarin, published in English, French, Russian, German, and Danish from 1936 to 2023 on Pubmed. As a first step, we searched using the keywords ‘acetylsalicylic acid, ‘salicylate’, and ‘salicylate therapy’, ‘aspirin’ OR ‘warfarin’, AND [micronutrient] AND ‘interaction’. As a second step, we searched using the keywords ‘absorption’, ‘transport’, ‘metabolism’, ‘excretion’ for ‘acetylsalicylic acid’ OR ‘aspirin’ OR ‘warfarin’ AND [micronutrient]. 

## 3. Defining the “Hidden Hunger” Essentiality of Micronutrients

Micronutrient deficiency is sometimes called the ‘hidden hunger’ [92], underlining the importance of micronutrients for health. The term ‘micronutrients‘ is generally used to define essential vitamins, minerals, and trace elements required to sustain basic physiologic functions [93]. Micronutrients are necessary as cofactors for vital enzymatic reactions [94]. Severe deficiencies might cause or aggravate clinical symptoms [36,95]. 

Vitamins are classified as water-soluble or fat-soluble. Water-soluble vitamins are thiamine (B1) [96,97,98,99], riboflavin (B2) [96,100,101], niacin (B3) [96], pantothenic acid (B5) [96], pyridoxine (B6) [102], biotin (B7) [103], folate (B9) [104,105], cobalamins (B12) [106], and ascorbic acid (C) [107,108,109]. Fat-soluble vitamins are retinol (A), α-, β-, γ- and δ-tocopherols/α-, β-, γ-, and δ-tocotrienols (E) [110,111,112], calciferols (D), and phylloquinone/menaquinone (K). Minerals [113] are calcium [114], phosphorous [115], magnesium [116], sodium [117], potassium [118], and chloride [119]. Trace elements [113] are iron [120,121], copper [122,123,124,125], zinc [126,127], selenium [42,128,129], and iodine [130]. This list is nonexhaustive and the essentiality of some nutrients is still debated [113]. The American Institute of Medicine published recommendations for dietary reference intake for thiamine (B1), riboflavin (B2), niacin (B3), vitamin B6, folate (B9), cobalamins as methylcobalamin, hydroxycobalamin, cyanocobalamin (B12) [131], pantothenic acid (B5), biotin (B7), and choline [132]. It also published recommendations of dietary reference intake for vitamin A, vitamin K, arsenic, boron, chromium, copper, iodine, iron, manganese, molybdenum, nickel, silicon, vanadium, and zinc [133], as well as recommendations for vitamin C, vitamin E, selenium, and carotenoids [134]. A list of 10 extra compounds was proposed as they ensure the proper function of longevity proteins, and a shortage of these compounds may result in age-associated chronic disease and cumulative insidious damages [135]. This list of associated dietary compounds includes taurine [136,137,138], ergothioneine [139,140,141], pyrroloquinoline quinone (PQQ) [142,143], queuine [144], carotenoids such as lutein [145], zeaxanthin [145], lycopene [116,146], α-carotene [145], β-carotene [145], β-cryptoxanthin [145,147,148], and astaxanthin [149,150]. Choline is also considered an essential nutrient [135,151,152]. Phenolics are equally considered as vital human dietary components [153]. Other important nutrients are needed to allow for proper biological function. For instance, two macronutrients, fatty acids [154,155,156] (some being essential) and dietary amino acids [157,158], are also critical for metabolism [159]. Choline will be excluded as its interactions relate to a dietary choline metabolite produced by the gut microbiota [160]. Phenolics will be excluded as their interactions are more within the field of drug–herb interactions [161]. Our review will consider exclusively ‘micronutrient’ in the broadest meaning possible, including vitamins, minerals, trace elements, and selected associated dietary compounds (taurine, ergothioneine, PQQ, queuine). 

## 4. ASA and DNIs

### 4.1. Reported ASA DNIs

#### 4.1.1. Water-Soluble Vitamins

##### Thiamine (B1)

*Increased thiamine urinary excretion.* Based on one human case report (five patients at therapeutic doses for rheumatic fever) [162], and two animal models [163,164], ASA may increase urinary excretion of thiamine (Table 1). Clinical relevance was discussed elsewhere, suggesting prophylactic thiamine supplementation [165]. A short period of ASA intake increased thiamin excretion, while long-term use decreased excretion. According to Cleland, this might be due to a loss of body stores during the previous period of increased excretion (Figure 2). 

##### Niacin (B3)

*ASA modulating prostaglandins to reduce niacin-flush.* Inhibitory drugs of cyclooxygenase reduce flushing by mediating the formation of prostaglandins [166,167,168], improving quality of life in the therapeutic supplementation of niacin [169]. The vascular factor of niacin-flushing is mediated by the hydroxycarboxylic acid receptor 2 (HCA2) and involves cyclooxygenase-mediated formation of prostaglandin D2 and E2 [170,171]. The mechanism was seen in humans (32 healthy subjects) receiving niacin and ASA (80 or 325 mg dose) on four separate visits that were at least 24 h apart [172] (Table 1). A higher dose did not provide additional benefits in another human study (42 healthy subjects, 650 mg ASA daily for four consecutive days) [173] (Figure 2).

##### Folate (B9)

*Multifactorial folate renal excretion.* Initially seen in patients with rheumatoid arthritis [174,175,176], ASA was found to interact with folate. Increased urinary excretion, significant fall in serum levels due to competition of binding sites on serum proteins [177,178,179], and interference with folate–oenzyme metabolism are three suspected mechanisms explaining the interaction [180]. ASA may slightly increase urinary excretion of folate based on one human case report (one healthy woman) [177]. Several reports indicate that 70% of patients with rheumatoid arthritis have decreased serum folate, with ASA-induced alterations in serum folate binding [179] (Table 1) (Figure 2).

*Fall in folate serum concentration*. In Lawrence et al., ASA induced a significant reversible fall in serum folate. This finding might be generalized to non-steroidal anti-inflammatory drugs but not to drugs having only antipyretic or analgesic properties [180,181]. One study challenges if the interaction applies to low-dose ASA as well [182]. Indeed, all studies finding interactions between ASA and folate had high intakes of ASA (for instance, 650 mg ASA every 4 h for 3 days [177]) (Figure 2). 

*Potentiating ASA through folate supplementation.* One article reviewed a mechanism that may potentiate ASA through high-dose folate in acute coronary syndromes and other diseases associated with increased platelet oxidative stress, namely modulation of nitric oxide synthase [183] (Figure 2).

##### Cobalamins (B12)

*Altered absorption of vitamin B12.* In a study on 255 patients with cardiovascular diseases, vitamin B12 deficiency was strongly related to the use of ASA [184]. This proposed mechanism could involve reduced absorption due to side effects of ASA on stomach mucosa and reduced secretion of the intrinsic factor [184,185,186,187]. The authors found a protective effect of Helicobacter Pylori on vitamin B12 absorption, supposedly from increased gastric acidity and an increased level of free vitamin B12. Generally, it is seen that Helicobacter is not protective but rather a cause of alterations. Helicobacter Pylori is generally seen as an important factor in the complex [188] cobalamins absorption [189]. Interestingly, unpublished data from a study on B vitamins for improved cognitive functioning in older people with mild cognitive impairment did not find any interaction between ASA and vitamin B12 [190]. Details on interactions of cyanocobalamin, methylcobalamin, and hydroxycobalamin with ASA are not available (Figure 2).

##### Ascorbic Acid (C)

*Unaffected absorption of ASA.* Through multi-spectra and voltametric studies, ASA was found to influence vitamin C binding with human serum albumin, but this influence does not affect the absorption of ASA [191].

*Decreased absorption of vitamin C* [22]. In animal models (guinea pigs and rats), ASA decreased gastrointestinal absorption of vitamin C [192,193], but the effect was thought to be different in humans due to the different absorption mechanisms between animals and humans [194]. A recent spectroscopy study revealed a mechanism that may reduce ascorbic acid absorption in concomitant administration of ASA and vitamin C [191]. Zhang et al. recommended against taking vitamin C concomitantly with ASA, since it was found to reducevitamin C absorption (Figure 2). 

*Decreased intragastric and serum vitamin C levels* [13,28,178,195]. In a double-blind study of 45 healthy human subjects, gastric mucosa and gastric juice was found to be among the largest depots of ascorbic acid, with 25 times the serum concentration (Table 1) [196]. A clinically high dose of ASA (800 mg ASA three times per day for 6 days) lowered ascorbic acid concentration by 10% in gastric mucosa. This decrease may be due to ASA-induced mucosal damage rather than decreased absorption [192], relating to the ascorbic acid gastroprotection effect in co-administration [13,197,198]. Whereas in the case of rheumatoid fever, high doses of ASA interact with ascorbic acid [195], the effect of chronic low-dose ASA on ascorbic acid status is unclear [13].

*Increased urinary excretion of vitamin C* [22]. The interaction was found in both animal (guinea pigs) [199,200] and human studies (case report of three children, 4–6 years old [201])—(Table 1). In humans, renal excretion stabilizes under chronic use of ASA, with a maximum excretion rate on the 6th day [202]. Enough ascorbic acid is retained for antiscorbutic effects, but there is a 114 percent impact on plasma and leukocytes concentration [199,202]. The derease in leukocyte ascorbic concentration was described in pediatrics as inhibition of vitamin C storage in leukocyte labile store [203]. Loh et al. therefore suggest prophylactic supplementation of ascorbic acid with chronic use of ASA [202] (Figure 2).

*Limited impact on specific neuronal Cox-2 inhibition* [204]. The neuroprotective effects of ascorbic acid might be related to a local Cox-2 inhibitory effect. Other studies found limited Cox-2 inhibition by ascorbic acid [205,206], but these findings cannot be extrapolated to overall Cox-2 [204].

#### 4.1.2. Fat-Soluble Vitamins

##### Tocopherols/Tocotrienols (E)

*Potentiation of antiplatelet effect with tocopherols and tocotrienols*. α-tocopherol has a mild antiplatelet and antioxidant effect [207,208,209,210]. While a sole α-tocopherol supplement is unlikely to have a clinically relevant antiplatelet effect [211], in conjunction with ASA, it might become relevant, increasing the risk of hemorrhage [212] or enhancing a preventive treatment regimen in patients with transient ischemic attacks [213]. Mixed tocopherols might be even more potent [214] (Figure 2). 

To harvest this cumulative mechanism and potentiate ASA, the concomitant use of tocopherols and tocotrienols was studied in a double-blind, randomized study on 100 patients (52 patients on ASA + vitamin E, 48 on ASA only) with transient ischemic attacks, minor strokes, or residual ischemic neurologic deficits, with highly significant reduction in platelet adhesiveness (2.7 × 10^5^ platelets adherent/cm^2^ for ASA + E–4.4 × 10^5^ platelets adherent/cm^2^ for ASA only) and significant reduction in the incidence of ischemic events in patients in the vitamin E plus aspirin group compared to the aspirin only group (three fatal events in ASA + E–6 in ASA only, one recurrent ischemic attack in ASA + E, two in ASA only, *p* < 0.05), with no significant difference in the incidence of hemorrhagic stroke between groups (three in both groups over 2 year period) (Table 1) [213]. High dose tocopherols (300 mg/day) are prone to antiplatelet interactions with ASA [215].

*Gastric protection against ASA-induced damage.* In an animal model (Sprague–Dawley rats), α-tocopherol showed gastric protection of the mucosa through suspected lipid peroxidation inhibition [216]. Tocotrienols are equally effective [217]. Another animal model study suggests that γ-tocopherol exhibits stronger lipid peroxidation inhibition than α-tocopherol and thus provides better gastric protection [218] (Figure 2).

#### 4.1.3. Minerals

##### Sodium

*Decreased excretion at high ASA dose.* Prostaglandins mediate renal sodium excretion and extracellular fluid volume regulation [219]. The effect may diminish the chronic use of nonsteroidal anti-inflammatory drugs [220]. Nevertheless, at high doses, decreased renal excretion might have a clinical impact on the patient (i.e., altered blood pressure) [221]. In adults, the dose-dependent renal effect of ASA may be considered from 80 mg per day [222] (Table 1) (Figure 2).

#### 4.1.4. Trace Elements

##### Iron

*Increased risk of anemia.* Using data from the Framingham study [223], Fleming et al. isolated data and correlated chronic ASA use (more than seven times a week) with lower serum ferritin levels (a decrease by 21 to 50%) [224] (Table 1), without being able to distinguish latent anemia from an anti-inflammatory response. Another cohort study in Denmark showed similar results [225] (Table 1). A recent study on the elderly using ASA found a fall in ferritin and an increased risk of anemia [226]. Another study analyzed the interaction of ASA and iron in cases of anemia without overt bleeding, with mixed results due to the quality of available data [227]. ASA is irritating to the gastric mucosa [62]. One commentary suggests using chronic ASA-induced iron loss as an anticancer mechanism [228]. It is also known that under certain conditions, ASA can bind with iron [229,230], even though no literature describes this in vivo (Figure 2).

#### 4.1.5. Associated Dietary Compounds

##### Taurine

*Booster of antiplatelet effect*. A human study (49 healthy adults, 24–45 years old, free of any medication for at least 15 days, no control group) found further inhibition of platelet aggregation with taurine supplementation [231] (Table 1). Taurine inhibits the aggregation induced by adenosine diphosphate [232]. Hence, coadministration of ASA and taurine might have some benefits in antithrombotic therapy [233], with a possible effect on ASA resistance [231].

*Possible gastric protection against ASA-induced damages.* In an animal model (Wistar albino rats, 220–300 g), taurine was hypothesized to protect the gastric mucosa [234].

*Alteration of excretion.* In a human study (six patients treated for rheumatic arthritis), ASA lowered urinary taurine excretion, evidenced by taurine clearance (urinary excretion divided by serum taurine concentration) and a mild rise in taurine serum levels. This mechanism might also be found in healthy subjects [235] (Table 1).
nutrients-16-00950-t001_Table 1Table 1Summary of clinically relevant DNIs with ASA. ↑ increase, ↓ decrease.NutrimentEffect on Nutrient Status or FunctionHuman StudiesReferencesNumberStudy DesignNumber of PatientsDosageResultthiamine (B1)↑ excretion1case report5647–1943 mg/day aspirin (ASA) for 5 daysmean urinary excretion thiamine ↑ 50%[154]niacin (B3)↓ flush (PGD2/PGE2 modulation)2Interventional—4 groups, 1 control31placebo—placebo;80 mg ASA—500 mg B3;325 mg ASA—500 mg B3;Placebo—500 mg B3↓ warmth,↓ flushing,↓ itching,↓ tingling,no difference between 325 mg and 650 mg ASA[164]interventional—3 groups, 1 control42Placebo—500 mg B3;325 mg ASA—500 mg B3;650 mg ASA—500 mg B3[165]folate (B9)↑ excretion3observational—37 patients, 59 controls37non indicated high-dose ASA (rhumatoid arthritis treatment)65% (24/37) subnormal folate serum level(<150 mcg/mL)[166]observational1650 mg ASA every 4 h, for 3 dayssubnormal serum folate level[169]observational182.1–3 g/day ASA, from 1 day to chronic usesubnormal serum folate level (<5 ng/mL)[171]cobalamins (B12)↓ absorption1descriptive cross-sectional observational study255low dose ASA for secondary prevention of ischemic heart disease①14% patients<150 pmol/L serum B12② 30% patients150–250 pmol/L serum B12[176]ascorbic acid (C)↓ C intragastric concentration1interventional—randomized, double-blind, parallel group453 × 80 mg ASA for 6 days↓ gastric mucosa concentration per 10%[188]↑ urinary excretion1case report3162 mg ASA 2 times at 3 days interval↑ urinary excretion[193]↓ C leukocyte concentration1interventional10600 mg ASA, 500 mg C↓ C leukocyte concentration by 114%[195]tocopherols/tocotrienols (E)antiplatelet potentiation of ASA1interventional100325 mg ASA and 400 IU α-tocopherol during 2 yearsplatelet adhesion reduced by 40% ASA + α-tocopherol group[205]sodium↓ urinary excretion1interventional—2 groups, 1 control16placebo and 160 mg ASA Group 1, 80 mg and 320 mg ASA Group 2interacting with ACE Inhibitors from 80 mg[214]iron↓ serum ferritin2Interventional—from Framingham heart study, 4 groups913number of ASA per week: non-user, 1–6, 7, >7↓ 25% serum ferritin from >7 ASA per week than non users[215]interventional—multiple cohorts170 on ASA arm; 1146 placebo armdose ASA unavailable—based on medical history↓Lower serum ferritin (median 136 mcg/L ASA, 169 mcg/L)[216]↑ anemia1interventional19,114placebo and 100 mg ASAincreased incidence of anemia and decline in ferritin[217]taurinepotentiation of ASA1interventional49400 mg and 1600 mg/day taurine for 14 daysdecreased aggregability through alteration in TXA2 release and GSH[223]↓ urinary excretion1interventional6high dose ASA for rhumatoid arthritisIncreased excretion of taurine[227]


### 4.2. Questionable ASA DNIs 

#### 4.2.1. Water-Soluble Vitamins

##### Riboflavin (B2)

*No known interaction.* In a study of the complexation of molecules (riboflavin, sodium salicylate, caffeine) in aqueous solution [236], it was found that a combination of riboflavin and sodium salicylate (an analogue of ASA) changed the biological activity of riboflavin. While the relation between caffeine and riboflavin is studied in the context of cutaneous melanoma [237], the relation between salicylates and riboflavin is currently unknown. In a human study (49 patients with migraine), 23 patients received 400 mg of riboflavin with 75 mg of ASA daily. No side effects were seen, except in one patient who withdrew from the study for gastric intolerance [238].

##### Niacin (B3)

*Niacin antiplatelet and fibrinogen effect.* Niacin has a direct inhibitory effect on platelet aggregation, corresponding with the metabolism of niacin in 12 human normal blood donors [239]. An hour after an oral dose of niacin, collagen and arachidonic acid-induced platelet aggregation showed a significant reduction, but not so after 12 h. In peripheral vascular disease, niacin has been shown to affect fibrinogen levels [240], which are highly correlated with changes in LDL-c [241]. The clinical significance thereof is unknown.

##### Pantothenic Acid (B5)

*No known interaction between ASA and pantothenic acid.* The antiplatelet effect of a biologically active intermediate is to be considered, although its clinical significance is unknown. 

*Pantethine antiplatelet effect.* Pantethine is a biologically active intermediate in the production of Coenzyme A, derived from pantothenic acid (vitamin B5). It is used as a supplement [242]. In a human study on 31 diabetic patients with hyperlipidemia, pantethine had mild antiplatelet aggregation properties [243].

#### 4.2.2. Fat-Soluble Vitamins

##### Retinol (A)

*No interaction in animal model*. A study [244] was done on 32 rats on the chronic administration of phenobarbitone, ASA (250 mg/kg/day), and oxytetracycline. The ASA group had normal vitamin A serum levels, comparable to that seen in the control group.

*Unclear gastric protection.* Several animal model studies suggested a cytoprotective effect for drug-induced gastric mucosal lesions [245,246,247]. 200 mg of ASA-induced gastric mucosal injury benefited from vitamin A administration [247]. But, in a double-blind, placebo-controlled study, no gastric protection was found with β-carotene [248], a provitamin A [249].

#### 4.2.3. Minerals

##### Phosphorous

*Increased urinary excretion.* The animal study on the calcemic and calciuric effect also monitored phosphorous excretion and discovered a significant increase from single dose ASA, but no alteration of excretion was found in chronic use [250].

##### Calcium

*Reduced serum and urinary calcium. A* study performed on normocalciuric rats showed reduced serum and urinary calcium after a single dose of ASA (−19.6% in serum), as well as with chronic ASA use (−20.8% in serum) [250]. The study suggests, by comparison with indometacin, two distinct mechanisms behind the observed effects–one being prostaglandin inhibition and the other remaining unidentified.

##### Magnesium

*Insignificant magnesium serum increase and decrease of magnesium urinary excretion.* Gomaa et al. [250] found an increase in magnesium levels, but it was insignificant compared to normal magnesium levels after a single dose of ASA. There was no alteration of magnesium excretion under chronic ASA. Magnesium bioavailability varies greatly depending on galenics [251,252].

##### Potassium

*No known interaction between ASA and potassium.* Conducting a PubMed search (55 results), no relevant studies appeared when the keywords used were ‘acetylsalicylic acid’, ‘aspirin’ and ‘potassium’ and ‘interaction’.

#### 4.2.4. Trace Elements

##### Copper

*Interaction between ASA and copper unclear.* Conducting a PubMed search (12 results), only Brumas et al. [253] appeared to be relevant when the keywords used were ‘ASA’, ‘ASA’ with ‘Copper’ and ‘Interaction’. ASA can bind with transition elements such as copper [229,230], and this interaction was found to potentiate the anti-inflammatory effect of ASA [254]. 

##### Zinc

*No clinically significant interaction between ASA and zinc.* In our PubMed search (19 results), no relevant studies appeared when the keywords used were ‘acetylsalicylic acid, ‘aspirin’ and ‘zinc’. ASA can bind with transition elements such as zinc [229,230]. 

##### Selenium

*Interaction unclear.* In a study on patients with ASA-sensitive asthma, lowered enzymatic activity of glutathione peroxidase activity was correlated with lowered serum selenium levels [255]. Inflammation is partially regulated by selenium and selenoproteins through the modulation of eicosanoid biosynthesis [256]. In an animal model, dietary selenium intake did not influence serum or renal excretion [257]. Selenium impacts the arachidonic acid cascade and inhibits the production of thromboxane A2 [258,259].

##### Chromium

*Increased absorption of chromium on rat model with ASA.* Two mechanisms are suspected: ASA can bind with transition elements, such as chromium [229,230], but increased absorption could also relate to competition for binding sites on ligands in intestinal absorption between chromium and other metals, like zinc and iron [260,261]. Another suspected mechanism is the inhibition of gastrointestinal prostaglandin synthesis [262].

#### 4.2.5. Associated Dietary Compounds

##### Lycopene

*Unclear additive antiplatelet effect.* Lycopene has an antithrombotic and antiplatelet effect [263,264,265]. One study compared platelet aggregation in vitro between ASA and concentrated lycopene and found a synergistic effect at 4 μmol/L of lycopene supplementation daily but not at higher concentrations [266]. For comparison, the median plasma lycopene in an observational study of plasma lycopene concentrations in 111 participants in a trial involving β-carotene was 0.59 μmol/L [267].

##### α-Carotene

*No known interaction between ASA and α-carotene.* In a PubMed search (five results), no relevant studies appeared when the keywords used were ‘acetylsalicylic acid, ‘aspirin’ and ‘alpha-carotene’. α-carotene is a provitamin A, contributing to 12–35% of newly converted vitamin A [268]. In an animal model, vitamin A reduced ASA-induced gastric injury [247], but a double-blind placebo-controlled trial on β-carotene, another provitamin A, showed no gastric protection [248].

##### β-Carotene

*No gastric protection against ASA-induced injury.* Beta-carotene is a provitamin A and a scavenger antioxidant. In a human double-blind placebo-controlled trial on the effect of chronic β-carotene supplementation (6 months) on the response to acute ASA injury (12 subjects, single oral dose of 650 mg of ASA dissolved in 60 mL of water), endoscopic control found mucosal lesions three hours post-administration in supplemented patients, as well as in the control group [248]. Vitamin A was found to be protective, but not β-carotene, as a provitamin A.

##### β-Cryptoxanthin

*No known interaction between ASA and β-cryptoxanthin.* PubMed yielded one irrelevant paper using the keywords ‘acetylsalicylic acid’, ‘aspirin’ with ‘beta-cryptoxanthin’. β-cryptoxanthin is a provitamin A [147,148]. In an animal model, ASA-induced gastric injury benefited from vitamin A [247], but a double-blind placebo-controlled trial on β-carotene, another provitamin A, showed no gastric protection [248].

##### Astaxanthin

*No interaction between ASA and astaxanthin.* To test a synthetic astaxanthin derivative, a human study tested the interaction between ASA and astaxanthin on 12 ASA-free subjects and eights subjects treated with ASA. The 25 tested biomarkers of platelet, coagulation, or fibrinolytic activity were unaffected by astaxanthin intake in both groups [269].

### 4.3. ASA and Metabolic Misuse of Micronutrients

*Metabolic misuse.* ASA is an electrophoretic uncoupler of mitochondrial oxidative phosphorylation [26]. By reducing the electric potential at the inner mitochondrial membrane below 200 mV (negative inside), it inhibits ATP synthase (complex V) and increases the accumulation of several-fold drugs inside the mitochondria, leading to alterations in processes in the mitochondria matrix [270]. This same membrane potential mechanism makes ASA induce cell death through modulation of the voltage-dependent anion channel [271]. A decrease in bioenergetic efficiency is double-faced: it may be beneficial to cellular function through the reduction of reactive oxidative species production and the balancing of nutrient availability [272], but it may also contribute to increasing mitochondrial dysfunction and toxicity [273], which couldaccount for a significant proportion of adverse effects of prescription drugs [26]. Micronutrients involved in bioenergetics. such as iron, zinc, biotin, vitamin B6, pantothenic acid, and copper [274], would then be required with an increased need—as mitochondria produce around 40 kg of ATP each day [270]. Compensatory mechanisms were revealed in animals [275], and inadequate micronutrient intake would result in further mitochondrial decay [39,274,276]. The clinical relevance of the mechanism in humans has yet to be assessed.

### 4.4. Unstudied DNIs with ASA

The following elements did not yield any results through PubMed, using ‘acetylsalicylic acid’ and ‘aspirin’ and the respective substance as keywords. Further searches adding ‘absorption’, ‘transport’, ‘metabolism’, or ‘excretion’ yielded no relevant results: biotin (B7)—22 results; chloride—61; sulfur—6; iodine—16; manganese—40; molybdenum—6; fluoride—53; arsenic—52; boron—14; nickel—22; silicon—69; vanadium—7. Interactions between ASA and these elements were not reported.

## 5. Warfarin and DNIs

### 5.1. Reported Warfarin DNIs

#### 5.1.1. Water-Soluble Vitamins

##### Niacin (B3)

*One case report on synergistic effect of warfarin and niacin.* Niacin has an impact on fibrinogen [240,241], plasminogen inhibitor type 1 (PAI-1) [277], and lipoprotein(a) [278]. One case report found a probable interaction between warfarin and one week‘s intake of high dose niacin (1000 mg) with critically elevated INR (12.3) [279] (Table 2). The case report found no previous occurrence in the literature but offered four hypothetical mechanisms leading to interactions, suggesting careful monitoring with supplementation (Figure 3).

##### Folate (B9)

*Secondary drug–diet interaction inducing folate deficiency.* Studies failed to find a relation between homocysteinemia and warfarin levels, but they suspect a drug–diet interaction. Green leafy vegetables are among the important sources of folate [280], as are strawberries [281], whole grain, egg yolk, liver, and citrus fruit [282]. While not being part of an appropriate clinical practice for patients under warfarin [283,284], 68% of patients in a retrospective cohort study involving 317 patients reported being advised to limit or avoid vitamin-K rich foods [285], such as green leafy vegetables. Avoidance of green leafy vegetables might cause a folate deficiency in most patients, as early as after 6 months of therapy with warfarin [286,287] (Table 2) (Figure 3).

*No association of serum folate levels with bleeding*. In a longitudinal cohort human study on 719 patients taking long-term warfarin, high homocysteine levels (≥17.2 μmol/L) and serum folate (≥11.9 nmol/L) were not associated with bleeding events, but both were associated with cardiovascular events (myocardial infarction, ischemic stroke, peripheral arterial emboli) [288] (Table 2). 

*Increased clearance of S-7-hydroxywarfarin but of no clinical relevance.* Consistent with the lack of association between serum folate levels and bleeding, folic acid supplementation increased clearance of S-7-hydroxywarfarin but did not translate into a clinical change; INR and warfarin dose change being not significant [289] (Table 2). 

##### Ascorbic Acid (C)

*Conflicting interaction between high-dose vitamin C and warfarin.* The literature is conflicting [290]. Case reports from the 1970s published in JAMA [291,292,293,294,295] initiated animal [296] and human [297,298] studies contradicting any relevant interaction. Still, case reports are published showing interactions between ascorbic acid and warfarin (with INR climbing from 1.1 to 15.4 within 2 days after the discontinuation of vitamin C) [299]. Despite a lack of data to support or refute the interaction, one study hypothesizes an interaction through a common dietary source of vitamin C and vitamin K [298] (Table 2) (Figure 3).

#### 5.1.2. Fat-Soluble Vitamins

##### Retinol (A)

*Possible interaction with high levels of vitamin A.* One case report on 13 patients under warfarin on interactions between warfarin and mango fruit [300] (Table 2) referenced a literature article on the interaction between vitamin A supplementation and warfarin [301]. One in vitro analysis of interactions between retinols and cytochrome P450 found that a high retinol dose could inhibit CYP2C19 [302], one of the CYPs responsible for the 7-hydroxylation of the warfarin R-isomer [72] (Figure 3).

##### Tocopherols/Tocotrienols (E)

*Potentiation of warfarin with vitamin E.* α-tocopherols has a known mild antiplatelet effect [207,208,209] and has been discussed to be at risk of interaction with warfarin [303]. It was initially found as non-interacting [304] on a double-blind study comprising 25 patients. In more recent studies, serum vitamin E levels were predictive of hemorrhagic events, especially in vitamin K-deficient patients (one interventional study on 12 cardiology patients receiving long-term warfarin, with mild-to-moderate prothrombin times (range, 16.0–21.5 s), and one retrospective observational study on 566 consecutive patients) [305,306] (Table 2). A high dose of tocopherols (300 mg/day) is related to interactions with warfarin [215]. Interactions between vitamin E and vitamin K activity are established, but the precise metabolic pathway is still under discussion [307] (Figure 3).

##### Calciferols (D)

*Vitamin D status and chronic warfarin therapy*. In a human study (40 patients with deep vein thrombosis or pulmonary embolism and vitamin D deficiency (<20 ng/mL), oral doses of 50,000 IU vitamin D3 per week for 8 weeks enhanced the anticoagulant effect of warfarin and reduced the maintenance dose requirement [308]. In another human study (89 subjects), only 25 subjects had a normal 25-hydroxyvitamin D level (over 30 ng/mL); 43 patients had an insufficient level (21–29 ng/mL), and 21 had a deficient level (<20 ng/mL). A weak significant association between these levels and the warfarin sensitivity index suggests an interaction [309] (Table 2). Early studies are finding a role of vitamin D in the pathogenesis of thrombosis through the modulation of tissue factor and/or production of cytokines [310]. While no study explicitly analyzed the role of ABCB1 protein in the drug–nutrient interaction between warfarin and D vitamin, it is notable that the ABCB1 protein contributes to vitamin D absorption [311], and its polymorphism has an impact on warfarin dose maintenance [85] (Figure 3). 

*Vitamin D status and increased risk of arterial calcification*. In an animal study, massive artery calcification appeared with high doses of vitamin D [312]. The study describes that high doses of vitamin D might elevate serum calcium levels, and serum calcium elevation might correlate with the onset of artery calcification. The study also considers the role of elevated phosphate serum levels. Warfarin inhibits the activity of the matrix Gla protein as a calcification inhibitor [40] (Figure 3).

##### K Vitamin

Warfarin is known for the inhibition of hepatic vitamin K epoxide reductase, inhibiting the reduction of vitamin K required for carboxylation of factors II, VII, IX, and X [64]. It is expected that this will decrease the gamma–carboxylation of 17 vitamin K-dependent proteins [313]. As reviewed by McCann and Ames [40], the then 14 known vitamin K-dependent proteins are categorized into three groups: four anticoagulation factors—prothrombin (factor II), factor VII, factor IX, and factor X—believed to all be mostly gamma–carboxylated in the liver; three anticoagulation regulatory proteins—protein C, protein S, and protein Z—mainly gamma–carboxylated in the liver, but also in other tissues; and Matrix GLA protein, osteocalcin, Gas6, Tgfbi, periostin, and proline-rich, Gla proteins 1–4. The report by McCann and Ames is one of the two demonstrating the triage theory in micronutrition [39,42]. Vitamin K is preferentially transported into the liver, where the most urgently required proteins are metabolized, related to coagulation functions [314]. Only when there is hepatic vitamin K sufficiency is vitamin K transported to extrahepatic tissues [315]. Lack of menaquinone-7-trans (the vitamin K2, MK-7-trans form) results in the inactivation of extrahepatic vitamin K-dependent proteins such as matrix GLA protein [316,317] and osteocalcin [318]. The role of the ABCB1 protein in K vitamin transintestinal efflux has been explored in animal models [319]. While the ABCB1 protein also interacts with warfarin [85], it is currently unknown if this has any clinical significance (Figure 3).

*Bone mineral density reduction*. Studies are not consistent in design, and the impact of antivitamin K on bone mineral density is controversial [320,321]. A pediatric study (70 children with chronic conditions requiring warfarin therapy) reports low bone mineral density (<2.0) in 13% of the patients. Body mass index (BMI) and growth hormone deficiency were identified as risk factors for bone mineral density reduction [322] (Table 2). In an adult study of patients with rheumatic valvular disease (70 patients with mechanical valve replacement), a marked reduction in BMD was found with long-term warfarin use [323] (Table 2). A systematic review and meta-analysis found an increased risk of fractures from warfarin therapy [324], consistent with the inhibition of osteocalcin, the most abundant non-collagenous protein in the bone [325] (Figure 3).

*Arterial calcification and atherosclerosis.* The use of warfarin is associated with an increase in systemic calcification, including arterial calcification [326], through the inhibition of matrix GLA protein. The etiology of vascular calcification is complex and involves several factors [327]. The long-term effects are debated and include valvular stenosis, stroke, and kidney disease [328]. Valvular and arterial calcification is a major challenge for children under antivitamin K, such as Fontan patients with high and intermediate thromboembolic risk [329]. One study suggested that the extract of Ginkgo Bilola EGB761 might be able to alleviate warfarin-induced aortic valve calcification [330]. This result cannot be transposed to all forms of Ginkgo Biloba, as literature reports have documented serious interactions [331]. Specific supplementation of menaquinone-7-trans (K2, MK7-trans) could specifically target Matrix GLA protein function [316,317], but it is impractical on a clinical level, as low doses of MK-7 supplementation significantly influenced INR values (with a mean value of INR dropping by 40%) [332] (Table 2) (Figure 3).

#### 5.1.3. Minerals

In a review of drug–herb and DNIs with warfarin, it is recommended to delay the administration of warfarin and minerals/trace elements by two hours due to the theorical risk of warfarin reducing absorption by binding [333].

##### Magnesium

*Serum magnesium as a factor in stabilizing INR.* In a human study (169 patients, 18–70 years old, under warfarin therapy), serum magnesium levels are a factor stabilizing INR (1.8 ± 0.2 mg/dL in non-stable INR group, 2.0 ± 0.1 mg/dL in stable INR group; *p* < 0.001) [334] (Table 2). Magnesium (II) is an important component of the coagulation cascade, influencing factor IX or VIII [335,336] (Figure 3).

##### Potassium

*Secondary drug–diet-nutrient interaction.* Low vitamin K foods can be high in potassium [337], which can be detrimental in cases of chronic kidney disease (CKD) [338], as those patients are at risk of hyperkalemia.

#### 5.1.4. Associated Dietary Compounds

##### Astaxanthin

*Reported interaction with astaxanthin supplementation*. One case report (one patient with ischemic stroke) assessed the probability of a relation between warfarin and the supplement as plausible [339] (Table 2). Astaxanthin mildly inhibits CYP2C19 [340] at a level unlikely to cause interaction. The mechanism is unclear (Figure 3).

**Table 2 nutrients-16-00950-t002:** Summary of clinically relevant DNIs with warfarin. ↑ increase.

Nutriment	Effect on Nutrient Status or Function	Human Studies					References
		Number	Study Design	Number of Patients	Dosage	Result	
niacin (B3)	synergistic effect	1	case report	1	2.5 mg warfarin/day + 1000 mg Niacin/day	INR jumped from 18 months stable INR 2.0–2.9 to 12.3 in a week	[271]
folate (B9)	no association with bleeding	1	longitudinal cohort	719	86% patients in INR 2.0–3.5	no association	[277]
	↑ clearance of S-7-hydroxywarfarin	1	interventional	24	5 mg/day B9 supplementation	non significant changes in dose and INR	[278]
	dietary-induced Folate deficiency	1	observational	114	dose unavailable	impaired folate status in as little as 6 months	[275]
ascorbic acid (C)	no interaction	8	observational	57	mean 3.3 mg/day warfarin	no significant INR change between C vitamin and warfarin dose	[287]
retinol (A)	possible interaction	2	case reports	13	daily mango intake from 1–6 in 2 days to one month	exact mechanism unknown, suspected A vitamin intake through mango	[289]
tocopherols/tocotrienols (E)	potentiate with vitamin E	3	observational	566	dose unavailable	higher serum E predictive of hemorrhagic events	[294]
calciferol (D)	low 25-OH D status	1	observational	89	INR 2.0–3.5 for 3 months from at least 3 consecutive visits	25/89 normal (>30 ng/mL); 43/89 subclinical deficiency (21–29 ng/mL); 21/89 clinical deficiency (<20 ng/mL)	[298]
K vitamin	vascular calcification	4 quoted in McCann/Ames				MGLA inhibition can cause vascular calcification	[313]
	bone density reduction—pediatrics	1	observational	70	more than a year on warfarin	13% of patients with BMD < 2.0	[309]
	bone density reduction—adults	1	observational	70	more than a year, 1.25–8.75 mg warfarin/day	Significant decrease in lumbar spine BMD	[310]
MK7	influence INR		interventional	18	10, 20, 45 mcg/day K2 MK-7	mean lowering of INR of 40% at 10 microg, 60% at 20 microg	[319]
magnesium	stabilize INR	1	observational—2 groups, stable and unstable INR	169	various	Mg significantly lower in unstable patients, most influential INR stabilization factor in the study	[321]
astaxanthin	influence INR	1	case report	1	warfarin 3 mg, astaxanthin 16 mg	INR jumped from 1.4 to 10.38. Probable relationship through the scale	[327]

### 5.2. Questionable Warfarin DNIs

#### 5.2.1. Water-Soluble Vitamins

##### Riboflavin (B2)

*No clinically relevant interaction between warfarin and riboflavin.* Flavoproteins are involved [341] in the enzymes required for the biosynthesis of the hepatic vitamin K-dependent clotting factors in an animal model [342]. But, flavin deficiency did not impact vitamin K 2,3-epoxide reductase activity, as the latter is not a flavoprotein, nor did it impact the inhibition of the vitamin K epoxide reductase complex by warfarin [342].

##### Cobalamins (B12)

*No interaction between serum cobalamins and warfarin intake*. In a human study of 114 patients under warfarin, after 6 months of treatment with warfarin, no significant variation in plasma vitamin B12 levels could be seen [287]. 

#### 5.2.2. Minerals

In a review of drug–herb and DNIs with warfarin, it is recommended to separate the administration of warfarin and minerals (and trace elements) by two hours due to the theorical risk of warfarin reducing absorption by binding [333].

##### Calcium

*No interaction with calcium urinary excretion.* In a human study (11 men on warfarin for at least 90 days), no difference in calcium urinary excretion was seen between the patients taking warfarin and the control group, as well as between people taking warfarin and people who had stopped warfarin, suggesting that there is no important role of a vitamin K-dependent mechanism in renal calcium excretion [343]. 

##### Magnesium

*Mixed results with respect to absorption by magnesium antacids.* In an in vitro model of drug interaction in the gut, absorption of warfarin was lowered by 20% in the presence of magnesium trisilicate [344]. In a human study (12 healthy subjects, 23–32 years old), magnesium hydroxide had no impact on the absorption of warfarin [345]. Different magnesium galenics have different bioavailabilities and may explain different study results [251,346].

##### Sodium

Clinically used warfarin is a salt [347] to eliminate trace impurities from the amorphous form of warfarin [348]. In a PubMed search (1148 results), no relevant studies appeared when the keywords were ‘warfarin’ with ‘sodium’.

#### 5.2.3. Trace Elements

In a review of drug–herb and DNIs with warfarin, it is recommended to separate the administration of warfarin and minerals (and trace elements) by two hours [333].

##### Iron

*Interaction between warfarin and iron unclear.* Stenton et al. mentioned a theorical risk of decreased absorption of warfarin and iron through ligand binding [333]. Through a PubMed search, no case reports of warfarin and iron interaction appeared. Like for ASA, iron status might be impacted by bleeding. This was the hypothesis raised by a human study on increased iron requirement in patients on hemodialysis and antiplatelet therapy or warfarin [349]. Conversely, iron deficiency might alter warfarin metabolization through cytochrome P450, a heme-containing enzyme [350,351].

#### 5.2.4. Associated Dietary Compounds

##### Lutein

*No known interaction between warfarin and lutein.* In one study reviewing carotenoids‘ potential of inhibition of cytochrome P450, lutein was found to have no inhibitory effect on studied CYPs [340].

##### Zeaxanthin

*No known interaction between warfarin and zeaxanthin.* Using PubMed, no relevant studies appeared when the keywords used were ‘warfarin’ with ‘zeaxanthin’. Zeaxanthin is a mild inhibitor of CYP3A4/5 [340], one of the CYPs responsible for the 10-hydroxylation of the warfarin R-isomer, but at a level unlikely to cause interaction.

##### Lycopene

*No known interaction between warfarin and lycopene. N*o study investigated any synergistic effect with warfarin.

##### β-Carotene

*No known interaction between warfarin and β-carotene.* By PubMed search, no relevant studies appeared when the keywords used were ‘warfarin’ with ‘β-carotene’. It is debated if high β-carotene levels can lower α-tocopherol levels [303], which hypothetically could alter platelet aggregation. 

##### β-Cryptoxanthin

*No known interaction between warfarin and* β-cryptoxanthin. In a PubMed search, no study appeared when the keywords used were ‘warfarin’ with ‘β-cryptoxanthin’. β-cryptoxanthin is a mild inhibitor of CYP2C8 [340], one of the CYPs responsible for the 7-hydroxylation of the warfarin R-isomer, but at a level unlikely to cause interactions.

### 5.3. Coenzyme Q10

Coenzyme Q10 is a critical component of the mitochondrial respiratory chain, transporting electrons to complex III in the electron transport chain [352]. It is also an important antioxidant [353,354]. It is, from a chemistry perspective, an analogue of vitamin K [355,356]. Through this analogy, it (in particular, the R-enantiomer [357]) may be interacting with warfarin through cytochrome P450 [358]. There are several case reports on the interaction between coenzyme Q10 and warfarin [359,360,361]. Nevertheless, a randomized, double-blind, placebo-controlled, cross-over trial found no clinical significance of interaction in the coenzyme Q10 supplement (100 mg daily) and warfarin therapy [362]. In a meta-analysis, anticoagulants were found to be associated with increased death from all causes [363]. Induced mitochondrial dysfunction caused by oral anticoagulation, such as warfarin interacting with coenzyme Q10, is questioned [364,365] (Figure 3).

### 5.4. Unstudied DNIs with Warfarin

With respect to the following elements, PubMed searches yielded no results using ‘warfarin’ and the element’s name. Further research adding ‘absorption’, ‘transport’, ‘metabolism’, or ‘excretion’ yielded no *relevant* results: thiamine (B1)—2 results; pantothenic acid (B5)—0; pyridoxine (B6)—5; biotin (B7)—16; chloride—14; sulfur—0; copper—16; zinc—23; selenium—8; iodine—73; chromium—16; manganese—0; molybdenum—0; fluoride—7; arsenic—12; boron—3; nickel—8; silicon—36; vanadium—1; taurine—3; ergothioneine—0; pyrroloquinoline quinone—0; queuine—0; α-carotene—0.

## 6. Discussion

The daily dietary intake of nutrients (from macronutrients (fat, protein carbohydrates), and micronutrients (vitamins, minerals, trace elements)) plays an essential role in human health. The daily consumption of nutrients is not uniform, displaying intra- and interindividual variation. Intraindividual variation reflects the temporal fluctuations in nutrient intake experienced by an individual, influenced by factors such as meals, timing, appetite, lifestyle adjustments, as well as biological ones (absorption, transport, metabolism, and excretion influenced by the physiology and the pathophysiology specific to each individual).

Polymorphisms affecting the absorption, transport, metabolism, and excretion of micronutrients such as vitamins, minerals, trace elements, and other cofactors play a crucial role in complex biochemistry processes. These interactions occur at various stages, from ingestion to the final reabsorption in the excretion system. Micronutrients, such as vitamins and minerals, often work synergistically or competitively with one another, impacting their absorption and bioavailability—the balance between copper/zinc/iron is a classic example [366]. Additionally, the presence of various dietary components, such as fiber, and macronutrients, such as carbohydrates, fats, or proteins, can further modulate the absorption of essential nutrients. Gut microbiota might also influence the occurrence of interactions through the modulation of drug efficiency [367,368,369]. Nutrition research has made significant discoveries, highlighting the complexity of nutrition and the importance of a well-formulated diet to ensure optimal nutrient absorption, transport, metabolism, excretion of nutrients, as well as non-nutrients required to optimize phase I and phase II metabolic processes [28], gut microbiome status, and overall health.

DNIs are a challenge for clinicians. Available data are limited and are mostly based on case reports or post-marketing observational studies [44]. Nevertheless, it is recognized that DNIs may influence health outcomes in vulnerable populations, such as elderly, obese, critically ill, transplant recipients, patients receiving enteral or parenteral nutrition, chronically diseased patients, or patients under polypharmacy [44]. Polypharmacy is generally defined as five medications for an adult [370] or two medications at the same time (for either more than a day or more than a month [371]) for an infant. In the elderly, the number of medications is associated with poorer nutritional status [372]. Polypharmacy increases the risk of drug–drug interactions, making the overall nutritional picture harder to understand, with its cumulative and synergistic effects [373]. Pediatric patients are to be equally considered [374,375], especially vulnerable groups—for example, patients with complex congenital heart disease, such as patients with Fontan circulation [40,376,377,378,379,380,381,382], on chronic polypharmacy. Warfarin-induced arterial calcification, amplified by growth and vitamin D, is challenging the widespread use of ASA and antivitamin K on the pediatric population [312,383]. Among the high-risk medications for DNIs is warfarin [384]. It is likely that drugs with a high risk of common drug–drug interactions are also of concern [44], including, among others, drugs with a low therapeutic range and drugs interacting with CYP450 [385]. 

Furthermore, when listing the limitations of DNI literature, clinical importance should be considered. Micronutritional deficiencies are common in western countries as well as developing countries [36,374]. DNIs and polypharmacy theoretically increase the risks of micronutritional deficiencies, enhancing the risk of adverse effect on chronically ill people with impaired nutritional status [9].

Not all micronutritional deficiencies have evident symptoms and remain undiagnosed [98,386]. Unclear symptoms of many micronutrient deficiencies [387] might hide behind the pathophysiology and course of the disease.

Making the global picture even more complex, not all micronutritional deficiencies have immediate pathophysiologic effects. In triage theory [39], it is suspected that the limited availability of micronutrients can cause an increase in chronic diseases. On an evolutionary premise, short-term survival for reproduction is favored over long-term health [37,38,388]. The triage theory has been demonstrated from a biochemical perspective and in animal models, with K vitamin-dependent proteins [40,314] and selenoproteins [42], where mild micronutritional deficiencies can cause insidious damage, accelerating the deterioration of chronic diseases [386]. The issue might be particularly concerning for the pediatric population with chronic illnesses, as this would lead to accelerated deterioration of the disease with micronutritional deficiencies, or conversely, slowing down the deterioration process [135]. Thus, it is important that clinicians pay more attention to nutritional status as part of a normal medical assessment of each patient.

Several approaches might improve the patient’s clinical outcome. 

From the pharmacist and clinician’s view, systematically considering the DNIs.

From the clinician’s perspective, stratifying the risk of interactions as part of the patient assessment or drug regimen review process, on an individual patient basis, as suggested by Boullata [44,389]. 

As a patient-centered approach, doing a micronutritional assessment as part of the medical examination for vulnerable populations, even using supplementation [390]. This can be done through medical history and methods of questioning (anamnesis) [391], as well as appropriate biomarker assessment on an individual patient basis [92].

When facing a possible DNI, clinician can use a drug-interaction probability scale designed to consider the causal relationship [392]. This scale is already in use by clinicians for the evaluation and reporting of DNIs [44,91,393,394] (Figure 4).

Karadima et al. suggested a major change [9]—a healthcare model using predictive, preventive, personalized medicine [395], taking advantage of omics technologies [396].

Through clinical [397,398,399] use of genomics [400,401], nutrigenetics [402], proteomics [398,403], metabolomics [404,405], and bioinformatics [406], clinicians may detect signs of diseases at a preventive stage, as well as stratify the risk of DNI [9]. Omics can be used clinically to assess the efficiency of micronutritional intervention [407,408] or the functionality of systems such as antioxidative defenses [409]. Omics could bring new insights into the importance of personalized therapeutic nutrition [410]. Drug–nutrient–genome interactions are being explored in cardiology [411]. Polymorphisms in the vitamin K epoxide reductase complex (VKOR) have been found to impact the interaction between warfarin and K vitamin through drug–nutrient–genome analysis [412]. 

## 7. Limitations

Even though ASA and warfarin are among the oldest and most prescribed drugs worldwide, the review showed clear limitations. 

Analyzing the DNIs demonstrated a lack of unified methodology and unified dose. Human cohort studies are relatively small. Some ASA–nutrient interaction studies’ designs were meant for rheumatic arthritis patients receiving high-dose ASA, which cannot be extrapolated to cardiovascular protective low-dose ASA. 

Several protocols were realized on animal models only, questioning if the effects can be extrapolated to humans. The difference in gastrointestinal absorption of vitamin C between guinea pigs and humans illustrates that this is not always the case. 

For a non-negligible amount of micronutrients (15 out of 45 for ASA; more than 30 out of 45 for warfarin, including all trace elements), we are currently missing data and thus cannot include or exclude a DNI. 

Interactions between retinoids/carotenoids and ASA were also barely explored. Sometimes, even proper diagnostic tests for marginal micronutritional deficiencies are unavailable, like in trace elements [113]. Moreover, not all polymorphisms of the main mechanisms behind antiplatelet and/or antithrombotic effects of ASA and warfarin have been identified. Additionally, polypharmacy could equally change DNIs, particularly with synergistic mechanisms. 

All these limitations do not detract from the intermediate objective of this review—paving the way for risk stratification for DNIs and resulting micronutritional deficiencies with the chronic use of ASA or warfarin. 

## 8. Conclusions

DNIs are reported occasionally in both warfarin and ASA users. A limitation of this assessment is the scarcity of available data. Potential DNIs could be managed through various strategies.

## Figures and Tables

**Figure 1 nutrients-16-00950-f001:**
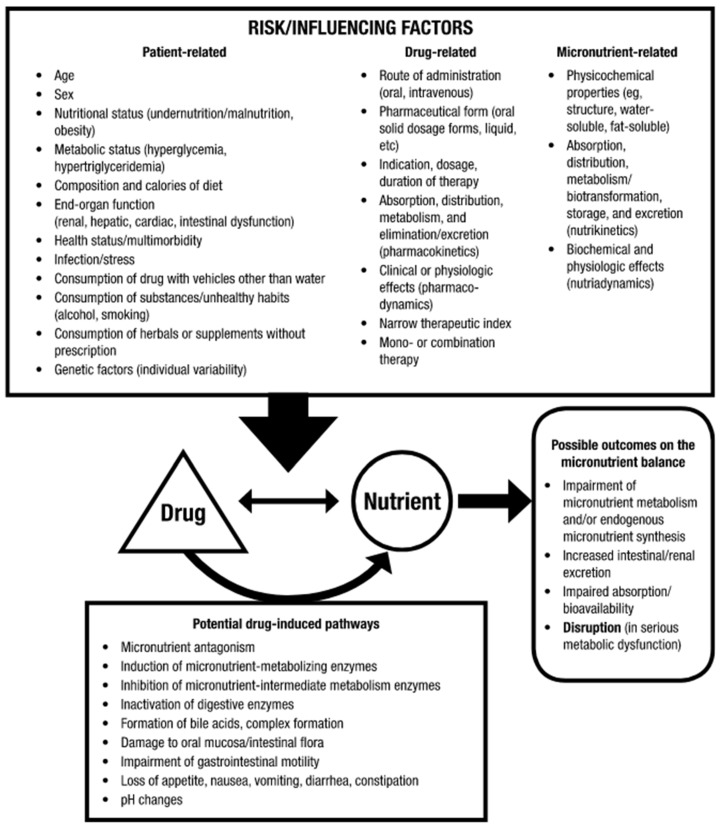
Bidirectional relationship of DNIs. Reproduced from Karadima et al. [9].

**Figure 2 nutrients-16-00950-f002:**
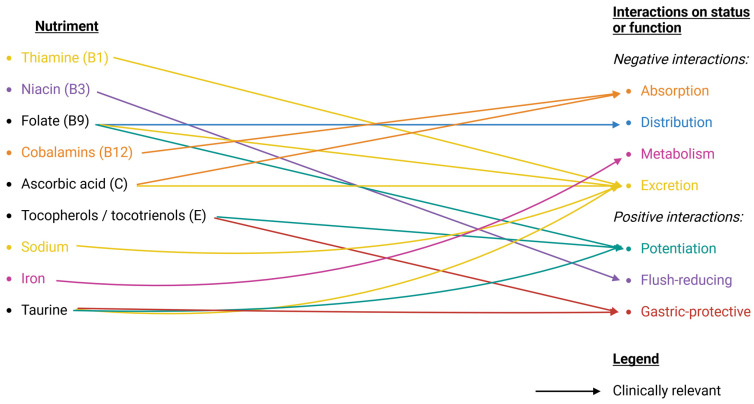
Selected DNIs with ASA. Created with biorender.com, accessed on 20 March 2024.

**Figure 3 nutrients-16-00950-f003:**
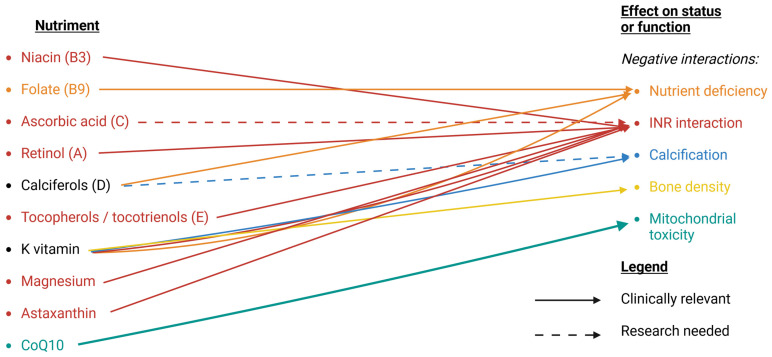
Selected DNIs with warfarin. Created with biorender.com, accessed on 20 March 2024.

**Figure 4 nutrients-16-00950-f004:**
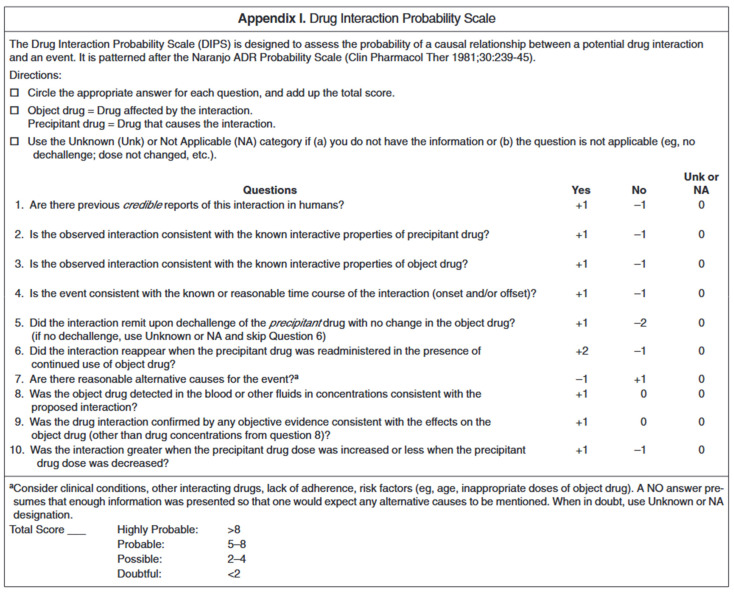
Drug Interaction Probability Scale. taken from Horn et al. [392].

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
