# Peer review of "Potential Drug–Nutrient Interactions of 45 Vitamins, Minerals, Trace Elements, and Associated Dietary Compounds with Acetylsalicylic Acid and Warfarin—A Review of the Literature"

_nutrients, 2024, doi:10.3390/nu16070950_

Round 1

Reviewer 1 Report

Comments and Suggestions for Authors

 In cardiology, acetylsalicylic acid (ASA) and warfarin are among the most widely used 13 prophylactic therapies against thromboembolic events. Chage to are commonly used.

Please, in its current form, figure 1 is quite confusing. Please, simplify. In addition, it is not clear what negative or positive effect means (i.e., negative effect on transport). The same for figure 2.

Table 1 is cut.

Methods should be placed after the introduction and the justification of the review.

Author Response

Dear reviewer,

In the name of all my coauthors, I would like to thank you very much for the time and effort you gave for our manuscript. We sincerely appreciate your insightful comments, which have significantly strengthened the quality of our work.

Comment #1: we indeed replace 'widely' by 'commonly' in the abstract.

Comment #2: you are right, in the current version, our two figures can be confusing the reader. While the difference between a negative and a positive drug-nutrient interaction is explained inside our text, it is not transparent in our figures. This is why we improved our figures with the new manuscript.

Comment #3: Table 1 cutting was a small editing mistake. Indentation of the table was too much to the right. We informed our editor and corrected it promptly.

Comment #4: You are also right, methodology should have been inserted between the introduction and the justification of our article. This is why it is now the case.

We have made the suggested revisions accordingly, and we believe these changes have improved the clarity and rigor of our manuscript.

Once again, we appreciate your thoughtful comments and constructive feedback. Should you have any further questions or require additional information, please do not hesitate to contact us.

Thank you for your consideration.

Warm regards,

David Renaud

Reviewer 2 Report

Comments and Suggestions for Authors

This review provides an overview of the relationship between aspirin and warfarin and vitamins, minerals and trace elements. The review covers a wide range of literature. My comments on the article:

1. The authors examine several members of the CYP enzyme family, but do not mention the transporter ABCB1, which interacts with them and is also very important in ADME processes. Please also examine the importance of this transporter for micronutrients with CYP involvement.

2. Metformin is known to enter cells through thiamine transporters, thereby reducing thiamine levels. ASA also reduces thiamine levels. Is it possible to know by which molecular mechanisms it does this? What happens to diabetics who use ASA and metformin together? Is there any data on this?

3. The situation is similar between ASA, folate and in the treatment of rheumatoid arthritis used methotrexate. Since rheumatoid arthritis is specifically mentioned, please elaborate on the ASA-methotrexate interaction.

4. The effect of warfarin and vitamin D on calcification raises the question of what other mechanisms may be responsible for this process. For example, how does pyrophosphate affect this process?

Author Response

nswer #2:

  Dear estimated reviewer,   In the name of all my coauthors, I would like to thank you very much for the time and effort you gave for our manuscript. We sincerely appreciate your insightful comments, which have significantly strengthened the quality of our work.   Overall comment: sadly, it will be hard to answer all your demands. Our review is limited to drug-nutrient interactions, and while it is a very important topic, drug-drug interactions are out of our range. I will cover this difference with more details answering your comments.   Comment #1: MDRP1 / ABCB1 role with CYP   ABCB1 is relatively poorly considered at this point in Drug-nutrient interaction review. At our knowledge, the current level of knowledge on MDRP1 in DNIs is of 3 levels:   - Multidrug resistance protein 1, as the clearest denomination, is involved into drug resistance through excreting cytotoxins out of the cell. Polymorphisms affect function of this protein and then of the drug efficiency. - the role of ABCB1 is poorly understood on nutrients. At our knowledge, only D and K vitamin with ABCB1 were explored.   We decided to include the role of ABCB1 in our article in 3 ways.   (1) Aspirin/Warfarin It was already mentionned in 1.3 under MDRP1.  In 1.4, we added the ABCB1 role for warfarin.   (2) Micronutrients At our knowledge, only the relationship between vitamin D and vitamin K and ABCB1 are currently explored. https://doi.org/10.1096/fj.201800956R - D vitamin https://doi.org/10.1016/j.foodchem.2020.128510 - K vitamin   In Aspirin, our review does not relate interactions with vitamin D and vitamin K (only vitamin A and vitamin E in fat soluble vitamins). But in warfarin, it does, and we included these two studies in our references.   (3) Regarding the influence of CYP for micronutrients, this is sadly the current limit of the literature. It is to be mentioned that only a small list of DNIs are documented, coming either from pharmacological studies, or from reports from clinicians. Pharmacogenomics would be an amazing addition to this topic. At our knowledge, no study analyzed DNIs between warfarin and vitamin D, vitamin K through ABCB1 polymorphism.   Comment #2: Metformin, Thiamine and Aspirin.   I understand your comment as two questions. The first question is the role of thiamine transporters with aspirin. The second question is the interaction between metformin and aspirin considering thiamine.   Question #1 Aspirin and thiamine transporters
  At our knowledge, ASA reduces thiamine through an unknown mechanism. I checked if there's any publication considering ASA and thiamine transporters.   Thiamine transporters are primarily two, but additional are existing (source: https://doi.org/10.3390/cells10102595). SLC19A1 SLC19A2 (ThTr1) SLC19A3 (ThTr2) SLC22A1 (OCT1) SLC25A19 (MTPP-1) SLC35F3 SLC44A4 (hTPPT/TPPT-1)   Through a pubmed search, I browsed them using the keywords we used to study DNI aspirin.   Here are the results:   OCT1 study found with Aspirin keyword is Mamlouk et al, also found with Salicylate keyword.   Details of the 6 studies found:     At the current stage of the literature, we can say that there is no proof of ASA interacting with thiamine transporters like metformin would do.   Question #2: interactions on thiamine with aspirin and metformin   The Pubmed search "aspirin" and "metformin" and "thiamin" gives no result. Nevertheless, polypharmacy is a real topic, increasing the risk of drug-drug interactions as well as micronutritional deficiencies. Whether aspirin and metformin both depleting thiamine might strengthen the risk of thiamine deficiency would require further research. Such studies exist for certain medications. Here an example on thiamine deficiency consecutive to furosemide and digoxin administration: https://doi.org/10.1016/s0014-2999(98)00710-9   We have carefully considered your recommendation on ASA and metformin considering thiamine. However, after thorough evaluation, we believe that implementing this modification may not align with the objectives and scope of our study. Our rationale for this decision stems from our study being focused on Aspirin and Warfarin, and not assessing drug-drug interactions having an impact on the micronutritional status of the patient. While it is indeed a significantly important topic for clinicians, it is out of the objectives and scope of our current article.   Comment #3: Aspirin and Methotrexate:
As for aspirin and methotrexate, polypharmacy is also a critical topic for DNI. Both medications are indeed related to a depletion in folate. I am not sure if it was deeply studied. Nevertheless, some results on pubmed search with 'Aspirin" and 'methotrexate' and 'folate'.  However, we feel that implementing an addition on ASA and methotrexate would not align with the objectives and scope of our study.   Comment #4: Warfarin, vitamin D and calcification:   The study of Price et al. (https://doi.org/10.1161/01.atv.20.2.317) on the increase of artery calcification with warfarin and high doses of D vitamin explains the mechanism - it is hypercalcemia:   'The vitamin D doses that cause artery calcification are also those that cause an elevation in serum calcium levels, and the time course of serum calcium elevation is correlated with the
onset of artery calcification.'   We acknowledged that it was not clear enough from your reading and we are suggesting the following modification: replacing ' The study considers the role of vitamin D in calcium homeostasis, as well as the elevated phosphate serum levels.' by   "The study describes that high doses of vitamin D might elevate serum calcium levels and serum calcium elevation might correlate with the onset of artery calcification. The study also consider the role of elevated phosphate serum levels."   ***   We understand the importance of addressing the concerns you raised and ensuring the robustness of our research. While we are unable to accommodate all your suggested modifications, we are open to discussing alternative approaches or solutions that may better align with the goals of our study.

Once again, we sincerely appreciate your valuable feedback and thoughtful suggestions. Should you have any further questions or require clarification on any other aspect of our manuscript, please do not hesitate to contact us. Your input is invaluable in enhancing the quality of our work.

Thank you for your understanding and consideration.

Best regards,

David Renaud

Round 2

Reviewer 2 Report

Comments and Suggestions for Authors

I accept the answers.